# cAMP Compartmentalisation in Human Myometrial Cells

**DOI:** 10.3390/cells12050718

**Published:** 2023-02-24

**Authors:** Alice Varley, Andreas Koschinski, Mark R. Johnson, Manuela Zaccolo

**Affiliations:** 1Department of Metabolism, Digestion and Reproduction, Imperial College London, Academic Department of Obstetrics & Gynaecology, Level 3, Chelsea & Westminster Hospital, 369 Fulham Road, London SW10 9NH, UK; 2Department of Physiology, Anatomy and Genetics, University of Oxford, Sherrington Building, Sherrington Road, Oxford OX1 3PT, UK; 3Oxford NIHR Biomedical Research Centre, Oxford OX4 2PG, UK

**Keywords:** cAMP, myometrium, pregnancy, phosphodiesterases, signalling compartmentalisation, hTERT-HM cells

## Abstract

Preterm birth is the leading cause of childhood mortality and morbidity. A better understanding of the processes that drive the onset of human labour is essential to reduce the adverse perinatal outcomes associated with dysfunctional labour. Beta-mimetics, which activate the myometrial cyclic adenosine monophosphate (cAMP) system, successfully delay preterm labour, suggesting a key role for cAMP in the control of myometrial contractility; however, the mechanisms underpinning this regulation are incompletely understood. Here we used genetically encoded cAMP reporters to investigate cAMP signalling in human myometrial smooth muscle cells at the subcellular level. We found significant differences in the dynamics of the cAMP response in the cytosol and at the plasmalemma upon stimulation with catecholamines or prostaglandins, indicating compartment-specific handling of cAMP signals. Our analysis uncovered significant disparities in the amplitude, kinetics, and regulation of cAMP signals in primary myometrial cells obtained from pregnant donors compared with a myometrial cell line and found marked response variability between donors. We also found that in vitro passaging of primary myometrial cells had a profound impact on cAMP signalling. Our findings highlight the importance of cell model choice and culture conditions when studying cAMP signalling in myometrial cells and we provide new insights into the spatial and temporal dynamics of cAMP in the human myometrium.

## 1. Introduction 

Preterm birth is the leading cause of mortality in children under the age of 5 years [1,2] and, due to its associated lifelong complications, it is a significant global health challenge [3,4,5]. Several complex molecular and cellular processes control the onset of spontaneous labour, but the precise mechanism is yet to be fully determined. As a result, there are no available interventions that can effectively stop labour once it has established at any stage of pregnancy.

Critical changes have been identified in the expression of key components of the cAMP signalling pathway during pregnancy and the transition to uterine activity with the onset of term and preterm labour [6,7,8,9,10,11,12]. A switch in the cAMP effector system has been characterised whereby, during pregnancy, cAMP activates protein kinase A (PKA), promoting a relaxed myometrial phenotype, whilst, with the onset of labour, cAMP acts via the exchange factor activated by cAMP (EPAC), triggering a contractile response [6,13]. The altered expression of different signalling components and the switch in cAMP effector activity at the onset of labour indicates the existence of different cAMP signalling subnetworks with distinct functions during the progression of pregnancy and is suggestive of compartmentalisation of cAMP in the human myometrium [14].

The development of fluorescence resonance energy transfer (FRET)-based reporters that are genetically encoded and can be targeted to specific subcellular compartments has helped elucidate the intricate spatial and temporal regulation of cAMP signalling in several cellular systems [14,15,16]. Using this approach it has been possible to analyse selectively cAMP networks organised at distinct subcellular sites and to establish the critical role of the cAMP-hydrolysing enzymes phosphodiesterases (PDEs) in the compartmentalisation of cAMP [14]. FRET-targeted reporters have facilitated the mapping of subcellular cAMP signals [17,18] and their association with specific G-protein coupled receptors (GPCRs) [19,20]. This has contributed to our current understanding of how multiple, diverse but highly integrated multi-protein complexes, or signalosomes, effectively coordinate the diverse functions mediated by cAMP, while maintaining hormonal specificity. Several studies have shown that disruption in individual signalosomes and of cAMP compartmentalisation is linked with the development of diseases [21,22,23], and manipulation of local cAMP levels has been proposed as a novel modality for therapeutic intervention with subcellular precision [24,25].

There is no information currently available on the subcellular compartmentalisation of cAMP signalling in myometrial cells and on its relevance for the physiology of pregnancy and labour. The myometrial cAMP system has previously been targeted in the management of preterm labour with the successful use of beta-mimetics causing uterine relaxation [26]. These drugs, however, cause concerning maternal side effects due to their systemic action [27], and safer, more effective, treatments are highly desired. 

Here we used FRET imaging to investigate cAMP signalling in human myometrial cells in pregnancy. Using targeted FRET reporters expressed in human primary myometrial cells (HPMCs) and in the myometrial cell line hTERT-HM, a cell model frequently used for studies on myometrial physiology [28,29,30,31], we uncovered the presence of compartmentalised cAMP signals triggered by catecholamines and prostaglandins at different subcellular locations. Our analysis also demonstrated that hTERT-HM cells and HPMCs display profound differences in cAMP signalling and that the number of passages in culture significantly affects cAMP signalling in HPMCs. We further observed that the cAMP response is highly variable in HPMC from different individuals.

## 2. Materials and Methods

### 2.1. Myometrial Tissue Collection 

Myometrial tissue biopsies (~0.5 × 0.5 × 0.5 cm) were taken from the upper portion of the uterine lower segment incision during Caesarean sections at Chelsea and Westminster Hospital between January 2018 and August 2019. Twenty women were included in the study who gave their written consent. Myometrial sample collection was conducted under the Preterm Labour (PREMS) study, which has approval from the Brompton and Harefield Research Ethics Committee (Reference number 10/H0801/45). The tissue samples were either snap frozen at −80 °C for extraction of mRNA and protein or stored in PBS at 4 °C for no longer than 3 h prior to preparation for cell culture. Inclusion criteria were as follows: singleton pregnancy, non-labouring, >37 weeks’ gestation, normal amniotic fluid levels, and preferably no more than two previous Caesarean sections. Caesarean section indications included fetal distress, previous Caesarean section, maternal request, or breech presentation. Patients were excluded if they had received an oxytocin infusion, or prostaglandins. See Appendix A for donor demographics. 

### 2.2. Primary Myometrial Cell Isolation 

Following collection, the myometrial biopsy was placed into PBS (Life Technologies Ltd. (Gibco), Paisley, UK, 14190169) at 4 °C for no longer than 3 h prior to preparation for cell culture. In a sterile laminar flow hood, the biopsy was rinsed with PBS several times to remove excess red blood cells, minced, and digested in a 50 mL tube containing a DMEM solution (Life Technologies Ltd. (Gibco), Paisley, UK, 41966052) with 0.25 mg/mL collagenase 1A (Sigma-Aldrich, Dorset, UK, CS674), 0.25 mg/mL collagenase XI (Sigma-Aldrich, Dorset, UK, C9891), and 5 mg/mL BSA (Thermo fisher, Hemel Hempstead, UK, 12676029) for 20 min in a 37 °C water bath whilst being agitated. Following this first digestion/agitation step, half of the resulting cell suspension was filtered through a 500 μm cell strainer (pluriSelect, Cambridge, UK 43-50500-03) and the individual cells were collected via centrifugation at 180× *g* for 5 min. The resulting supernatant was then returned to the remaining cell suspension and the digestion step was repeated for another 20 min. For larger biopsies, this process was extended for 5–10 min to ensure that the majority of the tissue was digested. The combined cell suspension was filtered again through a 500 μm cell strainer, and the cells were collected via centrifugation at 180× *g* for 5 min. The cell pellet was resuspended in DMEM containing 10% FBS (Life Technologies Ltd. (Gibco), Paisley, UK, 10500064), 100 units/mL penicillin, 100 µg/mL streptomycin (Sigma-Aldrich, Dorset, UK, P-0781) and 2 mM L-glutamine (Life Technologies Ltd. (Gibco), Paisley, UK, 42430025). On average, between 5 × 10^5^ cells to 7.5 × 10^5^ cells were collected per biopsy.

Depending on the cell yield and the experimental needs, cells were then seeded onto sterilised 15 mm diameter glass coverslips (VWR, Leicestershire, UK, 631-1579) in wells of 12-well plates (Corning Costar, Sigma-Aldrich, Dorset, UK, 3513) for FRET imaging experiments, or into 6-well plates (Corning Costar, Sigma-Aldrich, Dorset, UK, 3516) for RNA or protein extraction. Surplus cells were seeded into T75 flasks (Corning Costar, Sigma-Aldrich, Dorset, UK, 430641U) for subculture. HPMCs from 6 donors were used for quantitative PCR experiments. For western blotting, protein was extracted from HPMCs obtained from 7 donors. HPMCs were isolated from a total of 20 donors for FRET imaging experiments. 

All cells were kept in an incubator at 37 °C in a 5% CO_2_ humidified atmosphere for further experiments. The medium was refreshed every 2 days.

### 2.3. Protein Extraction

Protein samples were prepared from lysed primary myometrial cells grown in a monolayer using cell lysis buffer (1x Laemmli buffer, Bio-Rad, Hertfordshire, UK, 1610737) containing 50 mM DTT (Bio-Rad, Hertfordshire, UK, 1610611). The detached and lysed cells including the supernatant were stored at −80 °C to be used for western blotting. Protein concentrations of the individual samples were determined using the Bio-Rad Protein Assay (Bio-Rad, Hertfordshire, UK, 5000112) and a Bio-Rad iMARK plate reader (Bio-Rad, Hertfordshire, UK, 1681130) measuring the absorbances at 660 nm. Concentrations were then recalculated from a measured BSA standard curve.

### 2.4. Western Blotting 

The whole cell lysates in Laemmli buffer were heated to 96 °C for 5 min. Then, 20 μg of protein from each sample was loaded into each well on polyacrylamide gels (SDS PAGE precast gels, Bio-Rad, Hertfordshire, UK) alongside the Precision Plus Protein™ (Bio-Rad, Hertfordshire, UK, 1610374) pre-stained standard. The gels were subjected to SDS-PAGE electrophoresis and subsequently transferred onto polyvinylidene difluoride (PVDF) membranes (Rio-Rad, Hertfordshire, UK, 1704156) using the Trans-Blot^®^ Turbo^TM^ Transfer System (Bio-Rad, Hertfordshire, UK, 1704150EDU). The membranes were blocked with 5% *w/v* fat-free milk powder in 1X TBS-T solution for 1 h at room temperature and then hybridised with the respective primary antibody (Appendix A) overnight at 4 °C. The membranes were then washed for 1 h. Following this, they were incubated for 2 h at room temperature with the secondary antibody (Appendix A). ECL plus Clarity Western Substrate (Bio-Rad, Hertfordshire, UK, 1705060) were used for antibody detection and the membranes were imaged using an iBright™ FL1500 Imaging System (Invitrogen, Paisley, UK, A44241).

### 2.5. RNA Extraction

Total RNA was extracted and purified from primary myometrial cells grown in a monolayer using the RNeasy mini kit (Qiagen, Manchester, UK, 74004) as per manufacturer’s instructions. The concentration and purity of the RNA was quantified using a NanoDrop Nd-1000 spectrophotometer. The RNA was stored at −80 °C.

### 2.6. Quantitative RT-PCR 

Following quantification, 1.5 μg of RNA was reverse transcribed to cDNA with oligo DT random primers, PCR buffer, MgCl_2_ dNTPs, RNAse inhibitor, and MuLV reverse transcriptase using the QuantiTect Reverse Transcription kit (Qiagen, Manchester, UK, 205311). Primer sets were designed using the Primer 3 software and purchased from Invitrogen (Appendix A). A nucleotide Blast search was conducted to ensure the primer sequences corresponded to the gene of interest. Quantitative real-time SYBR Green PCR assays were performed with a RotorGene Q thermocycler using a pre-programmed sequence. The cycle threshold (Ct) was used for quantitative analysis, which denotes the cycle at which the fluorescence emission reaches a predetermined threshold. The cycle threshold was fixed at a level whereby the exponential increase in amplicon yield was approximately equivalent between the samples. A standard curve was used that involved a ten-fold dilution series. The mRNA concentration data were normalised to the amount of GAPDH-mRNA, which was used as the housekeeping gene and expressed as relative amounts.

### 2.7. hTERT-HM Cell Seeding and Infection 

A readily available hTERT myometrial cell (hTERT-HM) line was used [31]. The hTERT-HM cells were maintained in DMEM/Hams F12 (1:1) medium (Sigma-Aldrich, Dorset, UK D8327) supplemented with 5% charcoal stripped FBS (Life Technologies Ltd. (Gibco), Paisley, UK, 12676029), 100 units/mL penicillin, 100 µg/mL streptomycin, and 2 mM L-glutamine (refreshed every 2 days) at 37 °C in a 5% CO_2_ humidified incubator. At approximately 80% confluence, the hTERT-HM cells were passaged, and aliquots of 1.5 × 10^5^ cells in suspension were seeded onto sterile 24 mm diameter borosilicate glass coverslips (VWR, Leicestershire, UK, 631-1583) for FRET imaging experiments. The cells on the coverslips were then kept for 16–24 h under the same culture conditions as above. Subsequently, the cells were infected with viral vectors encoding for either a cytosolic FRET-based sensor (EPAC-S^H187^) [32] or a FRET sensor that was targeted to the plasma membrane via fusion to the scaffolding protein AKAP79 (AKAP79-CUTie) [33]. A total of 4.5 × 10^7^ virus particles of EPAC-S^H187^ or 3.8 × 10^7^ virus particles of AKAP79-CUTie sensor were added in solution directly to the culture medium of each coverslip. The hTERT-HM cells were incubated for 3 h with the virus after which the media was replaced, and the cells were further incubated 18–24 h prior to imaging of the EPAC-S^H187^ sensor or 48 h for the AKAP79-CUTie sensor. Efficiency of infection was 90–100% for the EPAC-S^H187^ sensor and 70–80% for the AKAP79-CUTie sensor. Cells were imaged at approximately 40–50% confluency.

### 2.8. HPMC Seeding and Adenoviral Infection

A total of 2.2 × 10^4^ cells were seeded onto 15 mm diameter sterilised borosilicate glass coverslips, which achieved a cell confluence of 30–40% for imaging of multiple cells per experiment, whilst assuring for a region devoid of cells to be used for background correction. 16–24 h after seeding, the coverslips were washed 2–3 times with PBS to remove excess red blood cells or residual tissue debris and the media was refreshed. Then, 4.5 × 10^7^ virus particles of EPAC-S^H187^ or 3.8 × 10^7^ virus particles of AKAP79-CUTie sensor were added in solution directly to the culture medium of each well. For both hTERT-HM cells and HPMCs, extensive trial experiments were performed to determine the optimum concentration of virus required for adequate sensor expression and infection efficiencies. A multiplicity of infection (MOI) of approximately 1000 virus particles per cell attained sufficient adenoviral transduction, which has previously been used in adult rat ventriculomyocytes [34]. Infection efficiencies of approximately 90% were achieved for the EPAC-S^H187^ sensor, and 70% for the cells expressing the AKAP79-CUTie sensor.

After infection, the coverslips were kept at 37 °C in a humidified 5% CO_2_ atmosphere for 18–24 h prior to imaging of the EPAC-S^H187^ sensor or 48 h for the AKAP79-CUTie sensor. The HPMCs were typically imaged between day 3 and day 6 after plating. For all experiments, except when testing for the effect of multiple passages in vitro, the cells were used without further passage.

### 2.9. Passage Experiments

Following isolation (deemed as passage 0 (P0)), HPMCs were grown to approximately 80% confluence in a T75 culture flask and sub-cultured for five subsequent passages (P1 to P5). At each passage, cells were seeded onto sterile 15 mm diameter borosilicate glass coverslips at a density of approximately 5.8 × 10^3^ cells/cm^2^ for FRET experiments. In conjunction with FRET imaging experiments, at each passage cells were also seeded into 6-well plates for RNA isolation. These cells were harvested when they reached about 80% confluency. The remaining HPMCs were subcultured into a new T75 flask. All cells were grown in DMEM media supplemented with 10% FBS, 100 units/mL penicillin, 100 µg/mL streptomycin, and 2 mM L-glutamine. Cell viability and growth was monitored microscopically every day.

### 2.10. FRET Imaging 

Sensitised FRET experiments were performed with an inverted Olympus IX71 microscope using a PlanApoN 40× NA 1.42 oil immersion objective, 0.17/FN 26.5 (Olympus, Southend-on-Sea, UK). Cells expressing the sensors were excited at a wavelength of 436 +/− 10 nm, and the excitation/emission dicroic mirror was 455 nm LP. An optical beam-splitter device (Dual-view simultaneous-imaging system, DV^2^ Mag Biosystems, Photometrics, AZ, USA) and a CoolSnap HQ^2^ monochrome camera (Photometrics, Tucson, AZ, USA) were used to record the YFP and CFP emissions in real time. The emission filter wavelength was 535 +/− 15 nm for YFP emission and 470 +/− 12 nm for CFP emission, with a beam-splitter dicroic mirror of 495 nm LP (Chroma Technology, Olching, Germany). MetaFluor, Meta Imaging Series 7.1 software (Molecular Devices, San Jose, CA, USA), was used for acquisition, storage, and offline analysis of the FRET data.

Ratio and FRET changes were calculated on background-corrected and, if applicable, drift-corrected emission intensities. These ratio- or FRET-change values correlated to changes in intracellular cAMP concentrations. Normalisation to the maximal response generated by forskolin (25 µM) and IBMX (100 µM) was conducted to allow for comparison of the two different FRET reporters used.

### 2.11. Statistical Analysis 

The distribution of data was determined using the Shapiro–Wilk test. Normally distributed data were analysed using a *t*-test (paired or unpaired). In cases of multiple comparisons, a one-way ANOVA followed by a mixed-effects analysis and Turkey’s multiple comparison post hoc test were used. Data that were not normally distributed were analysed using a Wilcoxon matched pairs test for paired data or a Mann–Whitney test for unpaired data. In cases of multiple comparisons, a Friedman’s test with a Dunn’s multiple comparisons post hoc test was used for paired data, or for unpaired data, a Kruskal–Wallis test followed by a Dunn’s multiple comparisons post hoc test.

Data were presented as the means +/− SEM. A value of *p* < 0.05 was considered statistically significant. The following symbol system was used to denote significance: * = < 0.05 < *p* < 0.01, ** = 0.01 < *p* < 0.001, *** = 0.001 < *p* < 0.0001, **** = *p* < 0.0001. GraphPad Prism 9.0 software was used to generate graphical representations of the data.

## 3. Results

### 3.1. Comparison of cAMP Responses in hTERT-HM Cells and HPMCs 

In this study we set out to compare the cAMP response to catecholamines and prostaglandins, two stimuli that play a key role in the regulation of the myometrium during pregnancy [35,36]. Our initial aim was to compare the response in two distinct subcellular compartments, the bulk cytosol, and the sub plasmalemma space of HPMCs, and in the cell line, hTERT-HM. To this aim, we employed two genetically encoded, FRET-based cAMP probes, the cytosolic Epac-S^H187^ sensor [32] and the plasmalemma-anchored AKAP79-CUTie sensor [33]. We first established that, when expressed in HPMCs, the Epac-S^H187^ (Figure 1A) and AKAP79-CUTie (Figure 1B) sensors show the expected localisation. Correct localisation of the sensors was also confirmed in hTERT-HM cells (not shown). To compare the cAMP response in the bulk cytosol and at the plasmalemma, cells expressing the sensors were treated with the *β*-AR agonist isoproterenol (ISO) or with prostaglandin E2 (PGE2), and the cAMP response was monitored by measuring FRET changes in the two compartments. The non-selective PDE inhibitor IBMX (100 μM) was applied to assess the contribution of the PDEs to the regulation of the cAMP response to the two agonists. The adenylyl cyclase (AC) activator forskolin (25 μM) was subsequently applied to achieve maximal cAMP generation and sensor saturation. 

As shown in Figure 2, while application of 1 nM ISO generated a robust cAMP response both in the cytosol (Figure 2A) and at the plasmalemma (Figure 2C) in hTERT-HM cells, no cAMP increase was detectable in either compartment in HPMCs (Figure 2A,C). Even at 1 μM ISO, the cAMP response remained lower at the plasmalemma and in the cytosol of HPMCs compared with the response elicited by 1 nM ISO in the two compartments of hTERT-HM cells (Figure 2A,C). In contrast, treatment with 1 μM PGE2 resulted in a significantly larger response both in the cytosol (Figure 2B) and at the plasmalemma (Figure 2D) of HPMCs compared with the response measured in the two compartments in hTERT-HM cells (Figure 2B,D), a difference that was maintained even when HPMCs were treated with 30 nM PGE2 (Figure 2B,D).

To investigate whether the differences between the two cell types may be due to a difference in local PDE hydrolytic activity, HPMCs and hTERT-HM cells expressing Epac-S^H187^ or AKAP79-CUTie were treated with ISO or PGE2 and subsequently exposed to the PDE inhibitor, IBMX (Figure 3). By subtracting the mean FRET responses observed after each agonist from the mean FRET change subsequently generated on application of IBMX, it was possible to assess the contribution of PDEs in regulating the cAMP levels achieved on agonist application. It is important to note that in no case was the FRET sensor saturated after addition of IBMX. The data show that, in hTERT-HM cells, the PDEs played a significantly different role in regulating the cAMP response to the two agonists. Specifically, we found that, both in the bulk cytosol and at the plasmalemma, the cAMP response to ISO was only minimally constrained by the PDEs, whereas the signal generated by PGE2 was significantly attenuated by PDE-dependent hydrolysis of cAMP (Figure 3A). In contrast, in HPMCs, the extent to which the PDEs constrained the cAMP response appeared to cover a wide range, resulting in an average value that was largely comparable for the two stimuli and the two compartments (Figure 3B).

Comparison of the kinetics of FRET change also showed differences in the cAMP response to ISO and PGE2 between cell types, and between the cytosol (Figure 4) and plasmalemma (Figure 5). In the cytosol, the cAMP signal measured over time in the continuous presence of an agonist shows a peak-plateau response with a relatively fast decline or, more often, an oscillating behaviour (Figure 4) that was particularly pronounced in HPMCs treated with PGE2 (Figure 4E,F). This oscillatory behaviour was never observed when the agonist was applied in combination with the PDE inhibitor IBMX (not shown). By contrast, the time course of the cAMP signal measured at the plasmalemma showed, in general, a sustained or slowly declining response (Figure 5).

### 3.2. Effects of Sub-Culture on HPMC Phenotype

Given the relatively limited availability of HPMCs, cell expansion through consecutive passages in culture is a common practice [37,38,39], although the possibility that such a procedure may impact on cell phenotype is recognised [40,41]. To assess whether in vitro passaging of HPMCs affects cAMP signalling, we set up subcultures from passage 0 (P0) to passage 5 (P5). At each passage, HPMCs expressing the cytosolic or plasmalemma sensor were treated with 1 µM ISO or 30 nM PGE2, followed by IBMX and forskolin, and the FRET changes recorded. We found that from P0 to P1 there was a significant increase in the cAMP response to 1 μM ISO detected in the cytosol (Figure 6A). This enhanced cAMP response was then sustained across subsequent passages (Figure 6A). A similar trend was observed in the plasmalemma, although in this compartment the difference between P0 and later passages did not reach statistical significance (Figure 6B). By contrast, no significant difference was observed in the cAMP signals generated by 30 nM PGE2 at either compartment across the different passages (Figure 6C,D). 

To investigate whether the larger response to ISO with passage was due to reduced PDE activity, we applied IBMX to the cells treated with the two agonists. As in previous experiments, in no case was the FRET sensor saturated after addition of IBMX.

As shown in Figure 7, the PDE contribution to the regulation of cytosolic cAMP signals generated by ISO declined with increasing passages, reaching statistical significance at P4 and P5 relative to P0. This was not the case at the plasmalemma, where the PDE activity was comparatively unchanged in subsequent passages (Figure 7B). For cells treated with PGE2, there was no significant difference in the relative contribution of PDEs across passages in either compartment (Figure 7C,D).

To investigate the cause of the enhanced cAMP response to ISO with increasing passages, we examined the gene expression profiles of PDE type 4B, *β*2-AR and EP2 receptor from P0 to P4. We found that, with increasing number of passages, the level of PDE4B gene expression significantly decreased by P4 compared with P0 (Figure 8A). Consistently, a decrease in the PDE4B protein level was also observed by P4, although the difference did not reach statistical significance (Figure 8B). The mRNA level for *β*2-AR (ADRB2) did not change with passage (Figure 8D), although a trend towards a decrease in protein level was observed (Figure 8E). There were no significant changes in the expression of EP2 receptor mRNA (PTGER2) (Figure 8G) or protein levels (Figure 8H) from P0 to P4. 

To further explore the effects of subculture on the HPMC phenotype, the expression level of the labour-associated genes connexin-43 (Cx43) and oxytocin receptor (OTR), and of the scaffold protein AKAP79, were also examined. We found no changes in the gene expression (GJA1) or protein levels of connexin-43 from P0 to P4 (Appendix A), while we found a significant increase in OTR gene expression (OXTR) (Appendix A). A significant increase in AKAP79 gene expression (AKAP5) was also observed with passage (Appendix A). The protein levels, however, were significantly decreased by P4 (Appendix A).

### 3.3. Variability of the cAMP Response in HPMCs from Different Donors

A striking variability in the cAMP response to agonist application both in the cytosol and at the plasmalemma was observed when analysing FRET changes in HPMCs obtained from individual donors (Figure 9). We found that, on application of 1 µM ISO, the cells from a small number of patients generated FRET changes of greater than 60% of the maximal response, whilst cells from other patients produced a much lower cAMP response (Figure 9A,B). Although the variability was particularly pronounced in responses to 1 µM ISO, a similar effect was evident also in cells treated with 30 nM PGE2 (Figure 9C,D). Notably, the amplitude of the cAMP response did not appear to be congruous in the two compartments or dependent on the specific agonist across different donors, with larger responses in the cytosol than at the plasmalemma in some donors or vice versa. 

## 4. Discussion 

The precise cellular processes involved in the initiation of spontaneous human labour have still not been fully determined. Solving this challenge would reduce the devastating complications and adverse perinatal outcomes of PTL [4]. A complex crosstalk of hormonal, biochemical, electrical, and mechanical influences are understood to activate and stimulate the myometrium to establish uterine contractions. Myometrial cAMP signalling is upregulated during pregnancy, promoting uterine quiescence [6,8,12,42,43,44]. Altered expression in its central signalling components and a switch in effector activity, in combination with the modulation of specific pro-labour genes such as OTR, are considered to promote the fundamental switch from a relaxed uterine state to a contractile one [6,8,12,13,45,46].

FRET imaging is a highly accurate and sensitive technique used to investigate, in real time, the cAMP response in specific subcellular compartments in living cells with unparalleled resolution in space and time [47]. In this study, we successfully expressed FRET-based reporters targeted at distinct subcellular sites in both hTERT-HM cells and HPMCs and established, for the first time, real-time imaging of cAMP in these cell models.

One objective was to determine if the hTERT-HM cells are a reliable model to investigate cAMP signalling in the myometrium in pregnancy. Our results demonstrate significant disparities between hTERT-HM cells and HPMCs in both the cAMP response to *β*-AR and EP receptor stimulation and in the regulation of the cAMP signal by PDEs. We found that ISO generates a significantly larger cAMP response both in the bulk cytosol and at the plasmalemma in hTERT-HM cells than in HPMCs. By contrast, activation of EP receptors elicits a significantly smaller response in hTERT-HM cells compared with HPMCs. 

Of the four EP receptors, EP2 and 4 couple with G*α*_s_, stimulate AC and promote uterine quiescence through increased cAMP production [48,49], while EP1 and 3 are associated with smooth muscle contraction, as they are coupled with Gα_q/11_ and Gα_I_, respectively, resulting in the activation of phospholipase C and the IP3/calcium pathway and inhibition of AC [50,51,52]. The hTERT-HM cell line was developed from non-pregnant myometrial tissue [53]. A study by Duckworth et al. examining the effects of butaprost, an EP2 receptor agonist, on both non-pregnant and pregnant myometrial tissue found that the inhibition of myometrial activity was greater in the pregnant tissue samples [54], suggesting that indeed the hTERT-HM cells may express lower levels of Gα_s_-coupled EP2 receptors compared with the HPMCs used in this study, which were obtained from pregnant women. Another consideration is that non-pregnant tissue biopsies are usually obtained from the anterior wall of the uterine fundus whereas pregnant samples are obtained from the lower uterine segment. Studies investigating EP receptor dominance in tissue from these different locations found that non-pregnant upper segment samples expressed mainly EP1 and 3 receptors, as opposed to lower segment pregnant tissue samples, where EP2 was predominant [55,56]. 

The finding of a larger cAMP response to ISO in the hTERT-HM cells compared with HPMCs may also be partly explained by differences in β2-receptor expression. Studies showed that in non-pregnant myometrial tissue the expression of β2-AR was approximately 50% higher, both at the mRNA and protein level, than in pregnant myometrial tissue [35,57]. Consistently, in functional studies, salbutamol was found to be more effective in inhibiting non-pregnant spontaneously contracting tissue strips than pregnant samples [35]. 

Analysis of the contribution of PDEs in determining the amplitude of the cAMP response to ISO and PGE2 suggests an additional mechanism that may account for the difference in cAMP responses observed between the two cell types. We found that application of IBMX to hTERT-HM cells resulted in a significantly larger cAMP increase in the presence of PGE2 than in the presence of ISO, indicating that the level of second messenger achieved in this cell line in response to PGE2 stimulation was markedly limited by PDE-dependent degradation, whereas the cAMP response to ISO stimulation was only minimally constrained by PDE activity. Thus, in hTERT-HM cells, a stronger coupling of PDE activity with EP2 receptors than with *β*-AR can explain, at least in part, the different cAMP levels observed in response to the two agonists. By contrast, in HPMCs, the extent to which the cAMP hydrolytic activity contributed in determining the cAMP response triggered by ISO or PGE2 appeared to be, on average, comparable for the two agonists, suggesting that, rather than a consequence of reduced degradation of cAMP by PDEs, the significantly larger response to PGE2 than to ISO observed in HPMCs may be due to more robust cAMP synthesis, consistent with higher expression levels of the EP receptors relative to β2-ARs in the HPMCs compared with the hTERT-HM cells. Further studies, however, are needed to confirm this hypothesis. 

Another important finding of the current study was the striking individual donor variability in the cAMP response in HPMCs. As the hTERT-HM cell line was generated from an individual donor [53], it is plausible that the differences observed between primary cells and the cell line reflected this individual variability, as exemplified by donor D41, which showed a response to agonist stimulation that was similar to the hTERT-HM cells. Inter-patient variability was demonstrated in contractility studies evaluating different β2 stimulants as potential tocolytics in term pregnant myometrial tissue strips [58]. A wide range of inhibitory effects to isoproterenol were observed across samples, which were correlated to the β2 receptor density [58]. Story et al. also detected variations in the inhibitory responses to isoproterenol in the relaxation of contracting term pregnant myometrial tissues, with 3 out of 11 samples exhibiting little or no effect [59]. The explanation for this insensitivity was not investigated but the authors speculated that it could be due to variations in receptor density or their affinity for the β2 agonist, altered GPCR coupling to Gα_s_/AC, or a reduction in AC activity [59]. Due to limited availability, there were differences in the number of cells analysed per donor. As a result, for certain donors, the *n* number was small, and this, to some extent, could have influenced the results on individual donor variability described above. However, significant variations were present also when comparing samples where we had the opportunity to image larger number of cells (e.g., compare D50 with D55, D57 and D58), supporting variation between individuals (see Figure 9 and Appendix A).

As mentioned above, a major limitation of using primary cells is their limited availability. In vitro expansion and re-plating is a commonly used procedure that allows an extended use of these cells, albeit at later passages after isolation. A study investigating the phenotype of HPMCs cultured for ten passages, from P1 to P10, found no difference in the structural morphology of the cells with passage [40] and no significant change in the total transcription of key smooth muscle markers, in the level of the labour-associated protein OTR, or in the response to inflammatory stimuli [40]. However, the study did not compare the cultured cells to freshly isolated HPMCs at P0, or to the tissue of origin. To date, the majority of studies have used HPMCs at lower passages, usually P4 or less [37,38,39]. Here we found that a significant increase in the response to ISO occurred from P0 to P1 in the bulk cytosol and, to a lesser extent, in the sub plasmalemma compartments, with substantially higher cAMP levels maintained at subsequent passages. One possible explanation for the larger cAMP signals produced to ISO with increasing passage could be due to an increase in β-AR expression. The protein levels of the β2-AR, which is the predominant receptor subtype in term pregnant myometrium [60], were investigated with passage. Surprisingly, the opposite was seen in that there was a significant reduction in receptor expression from P0 to P4. The reasons for this decrease are unclear and further studies will be required to confirm and explain these findings. Despite a reduced protein expression, mRNA levels for β2-AR remained unchanged with passages. This observation could be explained by increased protein turnover or effects due to altered efficiency of mRNA translation. Our results, however, do indicate that the enhanced cAMP response to catecholamine stimulation resulted, at least in part, from the progressive decline in PDE4B expression, which we documented both at the mRNA and protein level.

Interestingly, the reduced expression of PDE4B did not impact on the cAMP response to PGE2, indicating a preferential coupling of *β*-ARs with this PDE, and suggesting a compartmentalised regulation that resulted in a cAMP signal that was specific for each individual GPCR. The differential regulation of the cAMP signal generated by different agonists was also observed in hTERT-HM cells, where we found that the response to PGE2 was more tightly regulated by the hydrolytic activity of PDEs compared with the response to ISO. A second observation from this study confirmed compartmentalisation of cAMP in myometrial cells. We found a marked difference in the way the cAMP signal was handled in the bulk cytosol compared with the sub plasmalemma compartment. In the continuous presence of an agonist, the cAMP response was sustained in the sub plasmalemma compartment, whereas cAMP showed an oscillatory behaviour in the bulk cytosol, particularly in response to EP receptor activation. The cAMP oscillations appeared to be dependent on the activity of the PDEs. It should be noted that PDE4 isoforms are activated by PKA [61], and a plausible mechanism explaining the cAMP kinetics in the cytosol could involve the following steps: an increase in cAMP activates PKA which, in turn, phosphorylates and activates PDE4, resulting in cAMP hydrolysis; the PDE4 is then dephosphorylated, allowing the cAMP level to increase again and the cycle repeats, producing cAMP oscillations. Further investigations will be necessary to confirm this hypothesis and to establish the functional role of these cAMP oscillations and how they impact activation of downstream targets.

## 5. Conclusions

This study presents several novel findings concerning cAMP signalling in the human myometrium. The successful use of targeted FRET reporters uncovered the existence of compartmentalised cAMP signals in both HPMCs and hTERT-HM cells at distinct subcellular sites. Using this technique, we uncovered complex differences between the cell models in the cAMP pools generated to agonist stimulation, their unique regulation by PDEs and differential signalling kinetics in the cytosol compared with the plasmalemma. Marked changes were also identified in HPMCs after subculture, both in terms of amplitude of the cAMP response to specific agonist stimulation and in the expression level of key signalling markers. Overall, these data provide new insights into cAMP signalling in the human myometrium and set the foundation for future studies to define the role of compartment-specific cAMP signalling in pregnancy.

## Figures and Tables

**Figure 1 cells-12-00718-f001:**
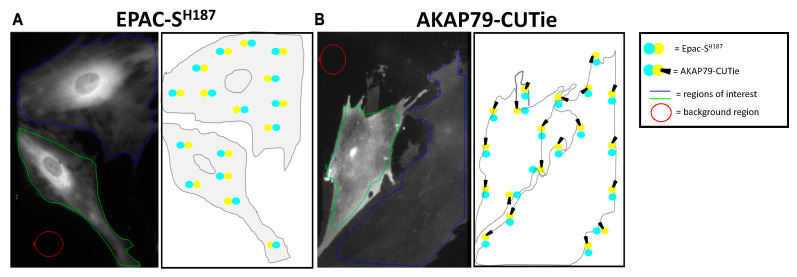
Localisation of EPAC-S^H187^ and AKAP79-CUTie sensors in HPMCs. YFP emission (shown in greyscale) and corresponding representative schematics demonstrating the localisation of EPAC-S^H187^ sensor (**A**) and AKAP79-CUTie sensor (**B**) in the HPMCs. For FRET measurements, grey values were averaged within regions of interest after subtraction of grey values averaged within the background region.

**Figure 2 cells-12-00718-f002:**
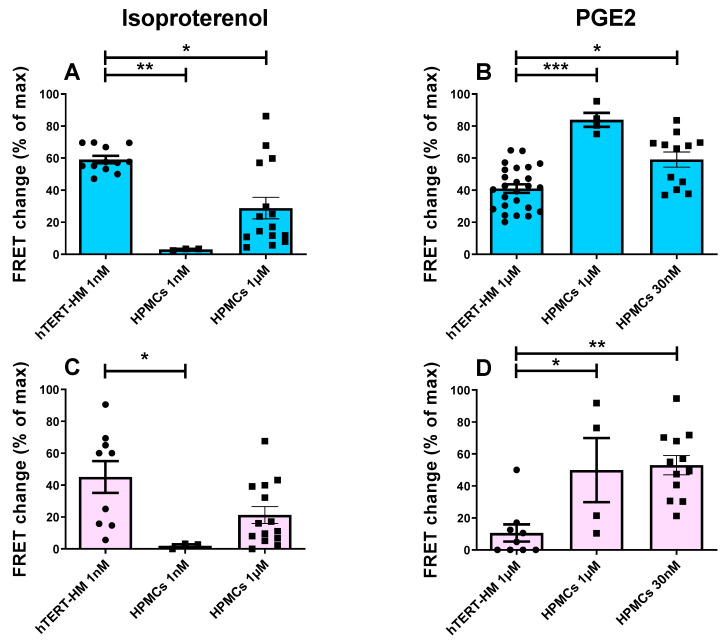
Comparison of the cAMP peak response in the hTERT-HM cells and HPMCs in the cytoplasm and plasmalemma following isoproterenol or PGE2 stimulation. FRET responses in hTERT-HM cells or HPMCs expressing the cytosolic (blue) Epac-S^H187^ sensor to (**A**) ISO treatment or (**B**) prostaglandin treatment. FRET responses in hTERT-HM cells or HPMCs expressing the plasmalemma (pink) AKAP79-CUTie sensor to (**C**) ISO treatment or (**D**) prostaglandin treatment. For hTERT-HM, each data point (circles) represents one cell (measurements from at least three independent cultures). For HPMCs, each data point (squares) indicates averaged measurements from individual donors; see Appendix A for cell numbers per donor. Data are expressed as changes relative to maximal FRET change at sensor saturation and show mean ± SEM; data normality was tested using the Shapiro–Wilk test followed by a Mann–Whitney test, * = 0.05 < *p* < 0.01, ** = 0.01 < *p* < 0.001, *** = 0.001 < *p* < 0.0001.

**Figure 3 cells-12-00718-f003:**
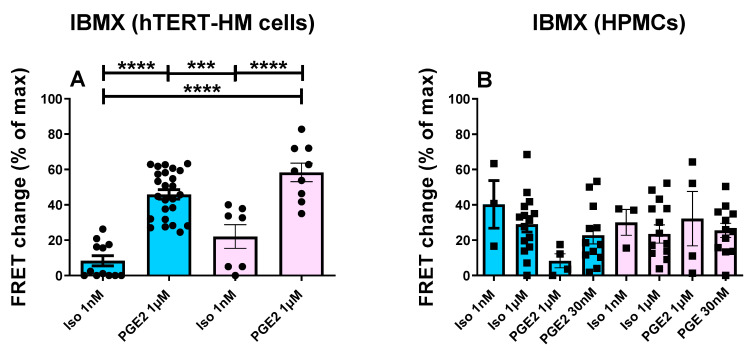
Comparison of the cAMP response to IBMX in the hTERT-HM cells and HPMCs in the cytoplasm and plasmalemma following agonist application. FRET responses to IBMX (100 μM) applied following ISO or PGE2 treatment of hTERT-HM cells (**A**) or HPMCs (**B**) expressing the Epac-S^H187^ sensor (blue) or AKAP79-CUTie sensor (pink). For hTERT-HM, each data point (circles) represents one cell (measurements from at least three independent cultures). For HPMCs, each data point (squares) indicates averaged measurements from individual donors; see Appendix A for cell numbers per donor. Values were calculated as the FRET change on application of IBMX relative to maximal FRET change at saturation and are presented as mean ± SEM; normality was tested using Shapiro–Wilk test followed by a Mann–Whitney test, *** = 0.001 < *p* < 0.0001, **** = *p* < 0.0001.

**Figure 4 cells-12-00718-f004:**
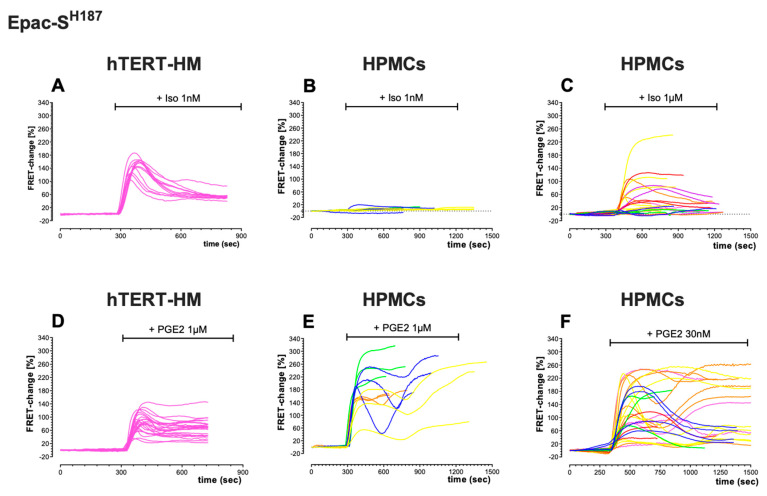
Kinetics of cAMP changes measured in the cytosol of hTERT-HM cells and HPMCs in response to ISO or PGE2 application. FRET-change kinetics recorded in individual cells expressing the cytosolic FRET sensor Epac-S^H187^. (**A**) hTERT-HM cells (*n* = 12 cells) treated with 1 nM ISO. (**B**) HPMCs (*n* = 11 cells, 3 donors) treated with1 nM ISO. (**C**) HPMCs (*n* = 26 cells, 7 donors) treated with 1 μM ISO. (**D**) hTERT-HM cells (*n* = 25 cells) treated with 1 μM PGE2. (**E**) HPMCs (*n* = 12 cells, 4 donors) treated with 1 μM PGE2. (**F**) HPMCs (*n* = 28 cells, 6 donors) treated with 30 nM PGE2. Same colour lines indicate cells from the same patient.

**Figure 5 cells-12-00718-f005:**
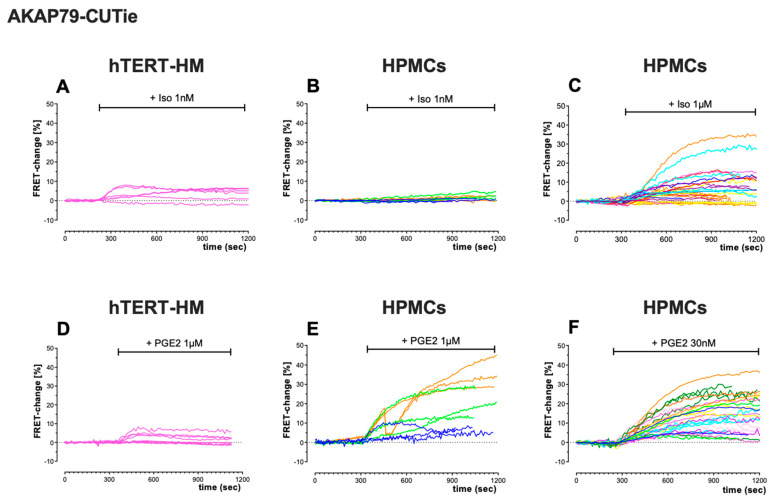
Kinetics of cAMP changes measured at the plasmalemma of hTERT-HM cells and HPMCs in response to ISO or PGE2 treatment. FRET-change kinetics recorded in individual cells expressing the plasmalemma-targeted sensor AKAP79-CUTie. (**A**) hTERT-HM cells (*n* = 6 cells) treated with 1 nM ISO. (**B**) HPMCs (*n* = 9 cells, 3 donors) treated with 1 nM ISO. (**C**) HPMCs (*n* = 29 cells, 7 donors) treated with 1 μM ISO. (**D**) hTERT-HM cells (*n* = 9 cells) treated with 1 μM PGE2. (**E**) HPMCs (*n* = 9 cells, 3 donors) treated with 1 μM PGE2. (**F**) HPMCs (*n* = 26 cells, 8 donors) treated with 30 nM PGE2. Same colour lines indicate cells from the same patient.

**Figure 6 cells-12-00718-f006:**
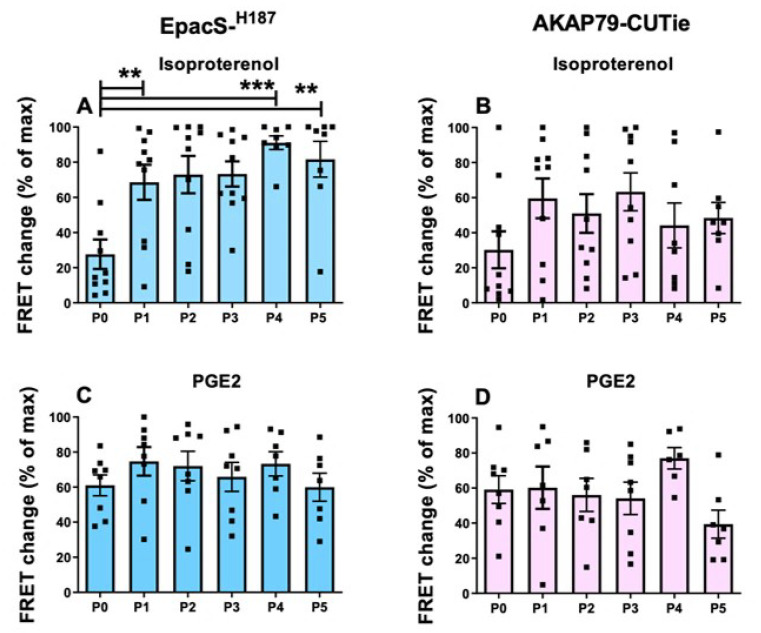
cAMP responses to ISO or PGE2 in HPMCs at passage 0 (P0) to passage 5 (P5) in the cytosol and plasmalemma. FRET responses to ISO (1 µM) in HPMCs expressing (**A**) Epac-S^H187^ (blue) sensor or (**B**) AKAP79-CUTie (pink) sensor and at different passages after isolation, as indicated. FRET responses to PGE2 (30 nM) in HPMCs expressing (**C**) Epac-S^H187^ (blue) sensor or (**D**) AKAP79-CUTie (pink) sensor and at different passages after isolation. Each data point (square) is the average of *n* = 3–5 cells per donor. Normality was tested using the Shapiro–Wilk test. For non-normally distributed data, a Kruskal–Wallis test followed by a Dunn’s multiple comparisons test was used, ** = 0.01 < *p* < 0.001, *** = 0.001 < *p* < 0.0001.

**Figure 7 cells-12-00718-f007:**
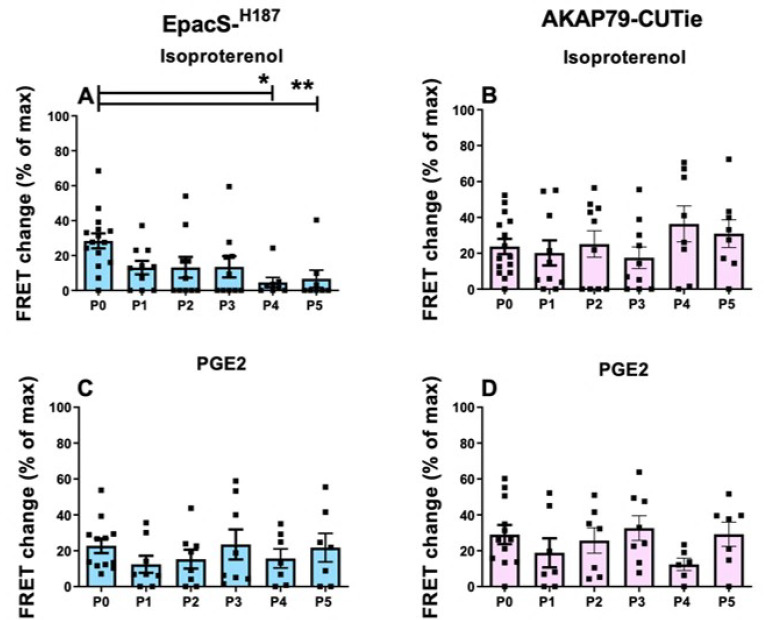
cAMP increase on inhibition of PDEs in HPMCs at passage 0 (P0) to passage 5 (P5) in the cytosol and plasmalemma and pre-treated with ISO or PGE2. FRET responses to IBMX (100 µM) measured in HPMCs pre-treated with ISO (1 µM) and expressing (**A**) Epac-S^H187^ (blue) sensor or (**B**) AKAP79-CUTie (pink) sensor and at different passages after isolation, as indicated. FRET change in response to IBMX (100 µM) measured in HPMCs pre-treated with PGE2 (30 nM) and expressing (**C**) Epac-S^H187^ (blue) sensor or (**D**) AKAP79-CUTie (pink) sensor and at different passages after isolation, as indicated. Each data point (square) is the average of *n* = 3–5 cells per donor. Normality was tested using the Shapiro–Wilk test. For non-normally distributed data, a Kruskal–Wallis test followed by a Dunn’s multiple comparisons test was used, * = 0.05 < *p* < 0.01, ** = 0.01 < *p* < 0.001.

**Figure 8 cells-12-00718-f008:**
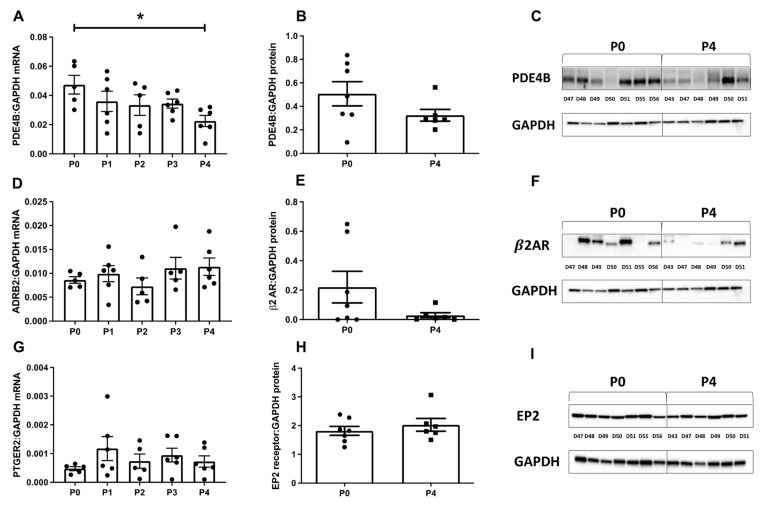
PDE4B, ADRB2 and PTGER2 mRNA expression and PDE4B, *β*2-AR and EP2 receptor protein levels in term no labour HPMCs from P0 to P4. HPMCs were cultured to 80% confluence at P0 and passaged to P4. At each passage, RNA was extracted and synthesised to cDNA for subsequent qPCR for PDE4B (**A**), ADRB2 (**D**) and PTGER2 (**G**). Protein was also extracted and analysed by western blotting at each passage. Densitometric analysis for PDE4B (**B**) with representative blot (**C**) *β*2-AR (**E**) with representative blot (**F**) and EP2 receptor (**H**) with representative blot (**I**). Data were normalised to GAPDH and show mean ± SEM. Normality was tested using the Shapiro–Wilk test. For mRNA, data were analysed using a one-way ANOVA followed by a Turkey’s multiple comparisons test. For protein, data were analysed using a paired *t*-test. * = 0.05 < *p* < 0.01. Each data point or protein band indicates HPMCs from individual donors. mRNA (circles) [*n* = 5–6]; protein p0 (circles) [*n* = 7]; protein p4 (squares) [*n* = 6].

**Figure 9 cells-12-00718-f009:**
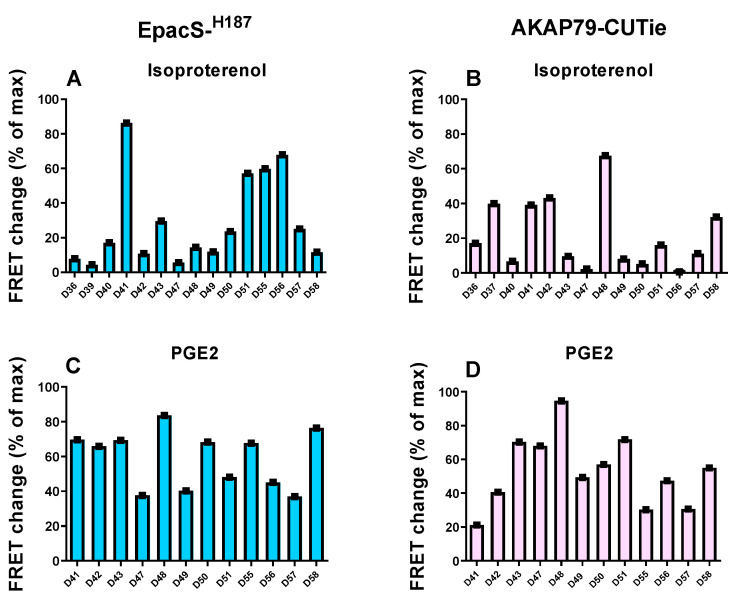
cAMP responses to ISO or PGE2 in the cytosol and at the plasmalemma of HPMCs from individual donors. Individual donor FRET response to ISO (1 µM) in HPMCs expressing Epac-S^H187^ (blue) sensor (**A**) or AKAP79-CUTie (pink) sensor (**B**). Individual donor FRET response to PGE2 (30 nM) in HPMCs expressing Epac-S^H187^ sensor (**C**) or AKAP79-CUTie sensor (**D**). For HPMCs, each data point (squares) indicates averaged measurements from individual donors; see Appendix A for cell numbers per donor. Data are expressed as changes relative to maximal FRET change at sensor saturation and show mean ± SEM.

## Data Availability

Not applicable.

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
