# Peer review of "cAMP Compartmentalisation in Human Myometrial Cells"

_cells, 2023, doi:10.3390/cells12050718_

Round 1
Reviewer 1 Report
Manuscript number: Cells-2227411
Title: AMP Compartmentalisation in Human Myometrial cells.
General comments
This paper expands information about experimental models and culture conditions for studying cAMP signaling in myometrial cells and provides new insights into the spatial and temporal dynamics of cAMP in the human myometrium. The experimental design is interesting, and the authors claimed that they found significant differences in the dynamics of the cAMP response in the cytosol and at the plasmalemma upon stimulation with catecholamines or prostaglandins, indicating compartment-specific handling of cAMP signals. In addition, their findings demonstrate significant disparities in the cAMP signaling in primary myometrial cells obtained from pregnant patients as compared to hTERT-infected myometrial cells and found marked response variability between patients. This manuscript sounds clear, and the findings are relevant to the physiology of pregnancy and labor. While very important and interesting the following needs addressing before it is considered for publication.
Detailed comments
Introduction
I strongly recommend the authors add the review article by Li et al (2021) https://www.ncbi.nlm.nih.gov/pmc/articles/PMC8106496/)
to the Introduction (lines 35-44) and Discussion sections of the manuscript since the dynamics of myometrial cAMP signaling, the role of EPAC, and crosstalk with other intracellular messengers were addressed by the authors in order to identify new ways to clinically modulate cAMP activity to reduce the occurrence of pregnancy/labor-related complications.
Methods
1. The authors should describe the suppliers of all reagents, culture media, and equipment cited in this section of the text.
2. Line 80: The authors should cite how many participants were included in the study and the time period during the myometrial tissue biopsies were collected (Please, see line 80).
3. Line 82: Please, cite the Research Ethics Committee Reference Number.
4. In order to improve readability and comprehension, please update the subheadings of the Methods section according to the order in which steps occurred. Can the following sequence make sense?
1. Myometrial tissue collection.
2. Primary myometrial cell isolation.
3. hTERT-HM cell seeding and infection.
4. Protein extraction.
5. Western blotting.
6. RNA extraction
7. Quantitative RT-PCR
8. HPMC seeding and adenoviral infection.
9. Passage experiments.
10. FRET imaging.
11. Statistical analysis.
5. Myometrial cell monolayer from how many patients were used for protein extraction, quantitative RT-PCR, and FRET analysis? I found this information difficult to understand in the figure subtitles and I suggest including it in the Methods section.
6. Lines 130-145: Is there a standardized protocol for optimizing hTERT-HM cell culture?
7. Could the authors comment on the differences in the virus particles of the EPAC-SH187 sensor added to the culture medium of hTERT-HM cells and HPMCs (5 x 107 vs. 4.5 x 107 virus particles, respectively)?
8. Lines 146-166: Were the isolated HPMCs assessed by Trypan blue exclusion method to determine cell viability before plating and culturing them?
9. 147-160: How many myometrial cells were obtained per biopsy on average?
10. The authors used DMEM plus 10% FBS or DMEM/Hams 1:1 supplemented with 5% charcoal FBS for primary myometrial or hTERT-HM cell culture, respectively. Besides the differences in the basal culture medium, the serum concentration is highly distinct between the two myometrial culture systems (HPMCs and hTERT-HMs). Why the HPMCs and hTERT-HM cells were not cultured in the same conditions with regard to the culture medium and supplements? It is important the authors discuss how these differences can impact or not the HPMCs and hTERT-HMs phenotypes and responses to the stimuli with agonists.
11. Line 153: After the first tissued digestion, the resulting cell suspension was filtered through a 500-μm-cell strainer. How was the final cell suspension obtained after the second tissued digestion procedure? Please, describe in the Methods the pore size and the manufacturer of the membrane filter used for the final cell dissociation from the primary tissue.
12. Line 159: Why did the authors culture the primary myometrial cells in DMEM medium supplemented with 10% FBS instead of the Smooth Muscle Medium containing 5% FBS? In addition, why did the authors not supplement the culture medium with antimycotic as Amphotericin B? Please, see Mosher et al, 2013 https://www.ncbi.nlm.nih.gov/pmc/articles/PMC3561148/)
13. Lines 209-210: To avoid redundancy, I recommend the authors exclude the sentences “for two groups“ (Line 209) and “for three or more paired groups” (Lines 209 and 210) when referring to the statistical tests performed (test t and one-way ANOVA, respectively). Please, instead of the sentence “For three or more paired groups” (Line 209) use the following sentence: “In cases of multiple comparisons” (…).
14. There is no mention of differences in cell number per patient between the treatment groups in the Methods and the Discussion sections. This data is important to be included so the reader understands whether variations in the cell number (For example, n= 3-18 per patient) can influence the quality and/or magnitude of the myometrial cells' responses to each stimulus applied.
Results
Please, check out all figures with regard p values description.
Page 7. Figure 2B: Were HPMCs treated with 1 µM and 30 µM PGE2 for cytosolic stimulation shown in Figure 2B? Please, add the PGE2 concentration.
Lines 278-286: Figure 3 is problematic since there is not any information on patients and cell numbers per patient in the legend. Also, the statistical significance was not shown in figure 3B. Check it out, please.
Line 327: To easily interpret the graphs in figure 6, please add the number of cell passages “(P1-P5)” to the figure title, or a description in the figure legend of what P1, P2, P3, P4, and P5 represent.
Lines 384-394: Could the authors provide a supplementary table showing the myometrial cell number per patient in figure 9?
Line 395-397: Please, clarify the abbreviation “AV” in the graphs of figure 9.
Discussion
Line 458: Could the authors comment if cell number has any influence on individual donor variability in the cAMP response in HPMCs assessment and how this contributed between treatment groups? Could it be a limitation of the study? May this, therefore, account for the variance if the cell number per sample was highly different as 3-18 per patient (Please, see line 343)? Or could it be due only to variations in receptor density or their affinity for the β2 agonist?
Line 461: Please, can the authors comment on how many myometrial cells were recovered for FRET analysis from patient AV41?
The authors should mention in the discussion advantages and disadvantages of the FRET technique.
Author Response
Response to Reviewer 1 Comments
Dear Reviewer,
Thank you for the considered reviews of our manuscript and we are delighted to enclose a revised version to address your comments.
Please see our responses and changes below, and we look forward to your comments.
Yours sincerely,
Best wishes
Alice Varley
Manuscript number: Cells-2227411
Title: cAMP Compartmentalisation in Human Myometrial cells.
General comments
This paper expands information about experimental models and culture conditions for studying cAMP signaling in myometrial cells and provides new insights into the spatial and temporal dynamics of cAMP in the human myometrium. The experimental design is interesting, and the authors claimed that they found significant differences in the dynamics of the cAMP response in the cytosol and at the plasmalemma upon stimulation with catecholamines or prostaglandins, indicating compartment-specific handling of cAMP signals. In addition, their findings demonstrate significant disparities in the cAMP signaling in primary myometrial cells obtained from pregnant patients as compared to hTERT-infected myometrial cells and found marked response variability between patients. This manuscript sounds clear, and the findings are relevant to the physiology of pregnancy and labor. While very important and interesting the following needs addressing before it is considered for publication.
Detailed comments
Introduction
I strongly recommend the authors add the review article by Li et al (2021) https://www.ncbi.nlm.nih.gov/pmc/articles/PMC8106496/) to the Introduction (lines 35-44) and Discussion sections of the manuscript since the dynamics of myometrial cAMP signaling, the role of EPAC, and crosstalk with other intracellular messengers were addressed by the authors in order to identify new ways to clinically modulate cAMP activity to reduce the occurrence of pregnancy/labor-related complications.
The suggested reference has been included in the manuscript (line 37 in the introduction and lines 427 and 430 in the discussion), thank you.
Methods
- The authors should describe the suppliers of all reagents, culture media, and equipment cited in this section of the text.
These have been included throughout the methods section where applicable.
- Line 80: The authors should cite how many participants were included in the study and the time period during the myometrial tissue biopsies were collected (Please, see line 80)
The text has now amended as follows:
' Myometrial tissue biopsies (~0.5 x 0.5 x 0.5 cm) were taken from the upper portion of the uterine lower segment incision during Caesarean sections at Chelsea and Westminster Hospital between January 2018 and August 2019. Twenty women were included in the study who gave their written consent'. (Lines 78-81).
- Line 82: Please, cite the Research Ethics Committee Reference Number.
This information has now been included as follows:
'Myometrial sample collection was conducted under The Preterm Labour (PREMS) Study which has approval from the Brompton and Harefield Research Ethics Committee (Reference number 10/H0801/45).' (Lines 81-83)
- In order to improve readability and comprehension, please update the subheadings of the Methods section according to the order in which steps occurred. Can the following sequence make sense?
- Myometrial tissue collection.
- Primary myometrial cell isolation.
- hTERT-HM cell seeding and infection.
- Protein extraction.
- Western blotting.
- RNA extraction
- Quantitative RT-PCR
- HPMC seeding and adenoviral infection.
- Passage experiments.
- FRET imaging.
- Statistical analysis.
The sequence has been altered in the methods section as per the suggestion of this reviewer. hTERT-HM cell seeding and infection has been moved to order number 7 above HPMC seeding and adenoviral infection due to further explanation of the viral infection method.
- Myometrial cell monolayer from how many patients were used for protein extraction, quantitative RT-PCR, and FRET analysis?I found this information difficult to understand in the figure subtitles and I suggest including it in the Methods section.
The number of patients used for qPCR, western blotting and FRET experiments have been listed more clearly in the methods section as follows:
'HPMCs from 6 donors were used for quantitative PCR experiments. For western blotting, protein was extracted from HPMCs obtained from 7 donors. HPMCs were isolated from a total of 20 donors for FRET imaging experiments.' (Lines 115-117)
- Lines 130-145: Is there a standardized protocol for optimizing hTERT-HM cell culture?
The protocol followed for hTERT-HM cell culture was recommended by the research group who provided the hTERT-HM cells for experimental study.
Please see reference 32, https://www.ncbi.nlm.nih.gov/pmc/articles/PMC3339884/).
- Could the authors comment on the differences in the virus particles of the EPAC-SH187 sensor added to the culture medium of hTERT-HM cells and HPMCs (5 x 107vs. 4.5 x 107 virus particles, respectively)?
Apologies, this is a typo mistake in the manuscript on line 172. This has been amended. For both cell types, 4.5x107 virus particles of EPAC-SH187 were used. Many thanks for pointing this out.
- Lines 146-166: Were the isolated HPMCs assessed by Trypan blue exclusion method to determine cell viability before plating and culturing them?
This method was not used to assess cell viability. An established protocol was used for myometrial cell isolation, see reference https://pubmed.ncbi.nlm.nih.gov/14742695/. Cell viability was assessed microscopically daily.
- 147-160: How many myometrial cells were obtained per biopsy on average?
Following extensive optimisation studies and adaptations to the cell culture methodology for HPMC isolation, on average between 5x105 cells to 7.5x105 cells were isolated per biopsy. This information has now been included on line 109.
- The authors used DMEM plus 10% FBS or DMEM/Hams 1:1 supplemented with 5% charcoal FBS for primary myometrial or hTERT-HM cell culture, respectively. Besides the differences in the basal culture medium, the serum concentration is highly distinct between the two myometrial culture systems (HPMCs and hTERT-HMs). Why the HPMCs and hTERT-HM cells were not cultured in the same conditions with regard to the culture medium and supplements? It is important the authors discuss how these differences can impact or not the HPMCs and hTERT-HMs phenotypes and responses to the stimuli with agonists.
The cell culture media and supplements used for the hTERT-HM cells were recommended by the research group who provided the hTERT-HM cells for experimental study. They have developed an established cell culture protocol for this cell type. As such, these were not altered for this reason.
Please see reference 32, https://www.ncbi.nlm.nih.gov/pmc/articles/PMC3339884/).
In addition, an established protocol utilised by our research group was used for primary myometrial cell isolation, see reference https://pubmed.ncbi.nlm.nih.gov/14742695/.
Prior to viral infection and subsequent FRET imaging experiments the cells require stable culture media conditions. Extensive experimental study has been conducted into the use of differing percentages of FBS and serum starved cells. Our research group has not found any significant differences between the different culture conditions with or without FBS in regard to variations in responses to agonist stimulation during FRET imaging experiments.
- Line 153: After the first tissued digestion, the resulting cell suspension was filtered through a 500-μm-cell strainer. How was the final cell suspension obtained after the second tissued digestion procedure? Please, describe in the Methods the pore size and the manufacturer of the membrane filter used for the final cell dissociation from the primary tissue.
Following optimisation studies, increased cell yield was noted following digestion of the minced tissue in a 37 °C water bath for two 20-minute intervals whilst continuously agitating the collagenase solution with a magnetic stirrer. After the first 20-minute agitation step, half the cell suspension was filtered through a 500μm cell strainer, and the cells in suspension collected by centrifugation. The media was returned to the flask and a second 20-minute agitation step was repeated to ensure the majority of the tissue was digested. This digestion step was halted following the addition of DMEM supplemented with 10% FBS. The total cell suspension was filtered again through a 500μm cell strainer, and the individual cells were collected via centrifugation. A 500μm mesh size cell strainer (pluriSelect, 43-50500-03) was used facilitating the solution to pass easily through the strainer, maximising the number of cells recovered, whilst ensuring larger tissue debris was retained on the sieve surface.
This information is now included in the manuscript on lines 92-109.
- Line 159: Why did the authors culture the primary myometrial cells in DMEM medium supplemented with 10% FBS instead of the Smooth Muscle Medium containing 5% FBS? In addition, why did the authors not supplement the culture medium with antimycotic as Amphotericin B? Please, see Mosher et al, 2013 https://www.ncbi.nlm.nih.gov/pmc/articles/PMC3561148/)
An established primary cell culture protocol was used which has been developed by our research group, see reference https://pubmed.ncbi.nlm.nih.gov/14742695/.
We ensure to conduct the cell culture process in optimum sterile conditions and have not previously experienced contamination of the HMPCs, therefore we do not routinely add antimycotics to our culture media.
- Lines 209-210: To avoid redundancy, I recommend the authors exclude the sentences “for two groups'' (Line 209) and “for three or more paired groups” (Lines 209 and 210) when referring to the statistical tests performed (test t and one-way ANOVA, respectively). Please, instead of the sentence “For three or more paired groups” (Line 209) use the following sentence: “In cases of multiple comparisons” (…).
The text has been revised as follows:
'The distribution of data was determined using the Shapiro-Wilk test. Normally distributed data were analysed using a t-test (paired or unpaired). In cases of multiple comparisons, a one-way ANOVA followed by a mixed-effects analysis and Turkey’s multiple comparison post-hoc test was used. Data which were not normally distributed were analysed using a Wilcoxon matched pairs test for paired data or a Mann-Whitney test for unpaired data. In cases of multiple comparisons, a Friedman’s test with a Dunn’s multiple comparisons post-hoc test was used for paired data, or for unpaired data, a Kruskals-Wallis test followed by a Dunn’s multiple comparisons post-hoc test.' (Lines 227-234).
- There is no mention of differences in cell number per patient between the treatment groups in the Methods and the Discussion sections. This data is important to be included so the reader understands whether variations in the cell number (For example, n= 3-18 per patient) can influence the quality and/or magnitude of the myometrial cells' responses to each stimulus applied.
We thank the reviewer for this suggestion, this is indeed an important point, and we now include this information in supplementary Fig 4 (see below).
|
Epac-SH187
(ISO 1nM)
Figure 2A, 3B & 9A |
Epac-SH187
(ISO 1mM)
Figure 2A, 3B & 9A |
Epac-SH187
(PGE2 1mM)
Figure 2B, 3B & 9C |
Epac-SH187
(PGE2 30nM)
Figure 2B, 3B & 9C |
AKAP79-CUTie (ISO 1nM)
Figure 2C, 3B & 9B |
AKAP79-CUTie (ISO 1mM)
Figure 2C, 3B & 9B |
AKAP79-CUTie (PGE2 1mM)
Figure 2D, 3B & 9D |
AKAP79-CUTie (PGE2 30nM)
Figure 2D, 3B & 9D |
|
D30 = 3 |
D36 = 3 |
D30 = 2 |
D41 = 5 |
D31 = 3 |
D36 = 3 |
D30 = 2 |
D41 = 2 |
|
D31 = 4 |
D39 = 3 |
D31 = 3 |
D42 = 3 |
D33 = 3 |
D37 = 4 |
D31 = 2 |
D42 = 3 |
|
D33 = 4 |
D40 = 4 |
D33 = 4 |
D43 = 3 |
D34 = 4 |
D40 = 4 |
D33 = 4 |
D43 = 2 |
|
|
D41 = 4 |
D34 = 4 |
D47 = 8 |
|
D41 = 3 |
D34 = 5 |
D47 = 2 |
|
|
D42 = 3 |
|
D48 = 6 |
|
D42 = 3 |
|
D48 = 3 |
|
|
D43 = 3 |
|
D49 = 5 |
|
D43 = 2 |
|
D49 = 3 |
|
|
D47 = 7 |
|
D50 = 12 |
|
D47 = 2 |
|
D50 = 6 |
|
|
D48 = 5 |
|
D51 = 3 |
|
D48 = 4 |
|
D51 = 3 |
|
|
D49 = 5 |
|
D55 = 7 |
|
D49 = 7 |
|
D55 = 2 |
|
|
D50 = 18 |
|
D56 = 5 |
|
D50 = 6 |
|
D56 = 8 |
|
|
D51 = 4 |
|
D57 = 9 |
|
D51 = 2 |
|
D57 = 5 |
|
|
D55 = 8 |
|
D58 = 10 |
|
D56 = 2 |
|
D58 = 4 |
|
|
D56 = 3 |
|
|
|
D57 = 4 |
|
|
|
|
D57 = 9 |
|
|
|
D58 = 10 |
|
|
|
|
D58 = 9 |
|
|
|
|
|
|
|
total = 3 donors |
total = 15 donors |
total = 4 donors |
total = 12 donors |
total = 3 donors |
total = 14 donors |
total = 4 donors |
total = 12 donors |
Table S4: Cell numbers per individual donor for the cAMP responses to ISO or PGE2 in the cytosol and at the plasmalemma of HPMCs.
We would like however to point out that it is unlikely that the patient variability that we find is the result of low number of cells that we have analysed for some patients, as such variability is also apparent for patients where we had the opportunity to image a larger number of cells (compare for example the responses in both compartments and both agonists detected in D50, with the response detected in D55, D57 and D58). We have added the following paragraph to the manuscript discussion to explain this:
'Due to limited availability, there were differences in the number of cells analysed per donor. As a result, for certain donors the n number is small, and this could to some extent influence the results on individual donor variability described above. However, significant variations are present also when comparing samples where we had the opportunity to image larger number of cells (e.g., compare D50 with D55, D57 and D58), supporting variation between individuals (see Figure 9 and Table S4).' (Lines 501-507)
Results
Please, check out all figures with regard p values description.
This has been amended where applicable in the results section.
Page 7. Figure 2B: Were HPMCs treated with 1 µM and 30 µM PGE2 for cytosolic stimulation shown in Figure 2B? Please, add the PGE2 concentration.
Figure 2B has been updated accordingly, thank you.
Lines 278-286: Figure 3 is problematic since there is not any information on patients and cell numbers per patient in the legend. Also, the statistical significance was not shown in figure 3B. Check it out, please.
In the figure legend we now refer to Suppl Fig 4 where we list the number of donors and number of cells per donor. There were no significant differences between the responses to IBMX for the HPMCs, for the two stimuli or between compartments in Figure 3B.
Line 327: To easily interpret the graphs in figure 6, please add the number of cell passages “(P1-P5)” to the figure title, or a description in the figure legend of what P1, P2, P3, P4, and P5 represent.
This has been added to the figure title of Figure 6. See below:
'Figure 6. cAMP responses to ISO or PGE2 in HPMCs at passage 0 (P0) to passage 5 (P5) in the cytosol and plasmalemma.' (Line 349)
'Figure 7. cAMP increase on inhibition of PDEs in HPMCs at passage 0 (P0) to passage 5 (P5) in the cytosol and plasmalemma and pre-treated with ISO or PGE2.' (Line 361)
Lines 384-394: Could the authors provide a supplementary table showing the myometrial cell number per patient in figure 9?
This has been included in the manuscript. Please see Table S4 in the supplementary material.
Line 395-397: Please, clarify the abbreviation “AV” in the graphs of figure 9.
For clarity, AV has been altered in the manuscript to 'D' for donor, which we define on line 419-420: ' D = donor.'
Discussion
Line 458: Could the authors comment if cell number has any influence on individual donor variability in the cAMP response in HPMCs assessment and how this contributed between treatment groups? Could it be a limitation of the study? May this, therefore, account for the variance if the cell number per sample was highly different as 3-18 per patient (Please, see line 343)? Or could it be due only to variations in receptor density or their affinity for the β2 agonist?
Please see our previous response to question 14. We do not feel that patient-to-patient variability is exclusively an effect of low cell numbers analysed for some donors, as it is apparent even for those individuals where we had the opportunity to analyse a larger number of cells. However, we acknowledge that one limitation of this study is that we were not able to confirm at the mRNA and protein expression levels the effects that we observed on activation of the ?2-AR using FRET imaging.
Line 461: Please, can the authors comment on how many myometrial cells were recovered for FRET analysis from patient AV41?
This has been included in the new supplementary table, Table S4. Please see our response to question 14.
The authors should mention in the discussion advantages and disadvantages of the FRET technique.
We describe to some extent the advantages of FRET in the introduction (see lines 46-60). This has now also been included in the discussion section of the manuscript. See below:
'FRET imaging is a highly accurate and sensitive technique used to investigate, in real time, the cAMP response in specific subcellular compartments in living cells with unparalleled resolution in space and time [48]' (Lines 432-434)
'Due to limited availability, there were differences in the number of cells analysed per donor. As a result, for certain donors the n number is small, and this could to some extent influence the results on individual donor variability described above. However, significant variations are present also when comparing samples where we had the opportunity to image larger number of cells (e.g., compare D50 with D55, D57 and D58), supporting variation between individuals (see Figure 9 and Table S4).' (Lines 501-507)
Reviewer 2 Report
Title: cAMP Compartmentalisation in Human Myometrial cells.
The role of cAMP in the control of uterine myometrial contraction has been widely studied, and some experiments have proved that cAMP is associated with delayed premature delivery. This manuscript further studies the compartmentalization of cAMP signals in myometrial cells, which helps us understand how multiple and diverse protein complexes or signal bodies can effectively coordinate different functions mediated by cAMP. This study provides new insights into subcellular precision therapeutic intervention. We found that the author used two cell models, in which HPMC cells were used for most of the experiments, and the primary cells without passage were selected to maintain the original characteristics of cells, making the results more reliable. In addition, the primary cells selected by the author are closer to the clinic and have great research significance. The overall experimental design is reasonable, clear and of high research value. Therefore, I suggest that this manuscript be accepted after some modifications.
Disadvantages:
1. The author selected EPAC-SH187 and AKAP 79-CUTie respectively in the localization of cytoplasm and plasma membrane, and the author can clarify the advantages of the two sensors.
2. In sections 2.6 and 2.8, when EPAC-SH187 virus and AKAP 79-CUTie virus infect hTERT-HM and HPMC cells respectively, how is the specific concentration of virus particles added determined?
3. We want to know how efficient the virus has been transduction to conduct the follow-up experiments? Could relevant evidence be provided to show that?
4. In the protein imprinting experiment, the author took GAPDH as the internal reference. Have you done similar experiments when the internal reference is β-Actin?
5. Please explain why there are seven bands in P0 generation cells and only six bands in P4 generation cells in Figure 8 C? Are the seven bands corresponding to P0 proteins extracted from seven different biopsy tissues?
6. The author lacks discussion on the phenomenon that the mRNA level from P0 to P4 b2-AR remains unchanged while the protein level has a downward trend as shown in Figure 8D, E and F.
7. Explain the meaning of red, green and blue borders in Figure 1.
8. In Figure 2 and Figure 3, it is suggested to add the meaning of triangle and square.
9. It needs more information about EPAC-SH187 and AKAP 79-CUTie. Like cloning or company.
10. In the discussion part, it is suggested to add and summarize the relevant mechanism of cAMP in playing the role of delaying premature delivery.
11. The tables mentioned in Line-105 and 119 are not found in the article.
Author Response
Response to Reviewer 2 Comments
Dear Reviewer,
Thank you for the considered reviews of our manuscript and we are delighted to enclose a revised version to address your comments.
Please see our responses and changes below, and we look forward to your comments.
Yours sincerely,
Best wishes
Alice Varley
Title: cAMP Compartmentalisation in Human Myometrial cells.
The role of cAMP in the control of uterine myometrial contraction has been widely studied, and some experiments have proved that cAMP is associated with delayed premature delivery. This manuscript further studies the compartmentalization of cAMP signals in myometrial cells, which helps us understand how multiple and diverse protein complexes or signal bodies can effectively coordinate different functions mediated by cAMP. This study provides new insights into subcellular precision therapeutic intervention. We found that the author used two cell models, in which HPMC cells were used for most of the experiments, and the primary cells without passage were selected to maintain the original characteristics of cells, making the results more reliable. In addition, the primary cells selected by the author are closer to the clinic and have great research significance. The overall experimental design is reasonable, clear and of high research value. Therefore, I suggest that this manuscript be accepted after some modifications.
Disadvantages:
- The author selected EPAC-SH187and AKAP 79-CUTie respectively in the localization of cytoplasm and plasma membrane, and the author can clarify the advantages of the two sensors.
The EpacS-H187 sensor was used to examine the global cytosolic cAMP compartment. It is a 4th generation sensor, which is constructed using the full length of EPAC protein and the fluorophores, mTurquoise2, a cyan fluorescent variant and a tandem of the 173 circularly permuted form of Venus (cp173Venus), a yellow fluorescent protein. For this sensor, the membrane targeting DEP-sequence has been deleted rendering it catalytically inactive in order to prevent interactions with downstream effector proteins. The use of mTurquoise, which has a higher quantum yield, photostability and low signal to noise ratio resulted in greater dynamic FRET ranges for this sensor. Circular permutation of Venus protein in tandem also increased the brightness of the acceptor channel and was found to be more resistant to UV and pH changes. These fluorophores significantly improved the FRET efficiencies of this sensor due to an increased affinity to cAMP and faster activation kinetics.
The AKAP79-CUTie sensor is part of a family of recently developed FRET-based probes which can directly target macromolecular complexes at distinct subcellular compartments via fusion to their individual protein components. This cAMP sensor is targeted to the plasma membrane via fusion to the scaffold protein, AKAP79. These sensors were designed using novel topology so that that there were limited effects on the FRET probe properties through minimising steric hinderance. This was achieved by moving the YFP from the N- terminus to loop 4 - 5 of the second CNBD of the regulatory subunit of PKA type IIb, the cAMP sensing moiety and inserting the targeting domain in its place.
As both these sensors have been described before, we now include the appropriate references https://journals.plos.org/plosone/article?id=10.1371/journal.pone.0122513 and https://www.nature.com/articles/ncomms15031 ) in the text.
- In sections 2.6 and 2.8, when EPAC-SH187virus and AKAP 79-CUTie virus infect hTERT-HM and HPMC cells respectively, how is the specific concentration of virus particles added determined?
Extensive trial experiments were conducted to determine the optimum concentration of virus needed to successfully achieve adequate expression levels of the targeted FRET biosensors in both myometrial cell types. The introduction of viral vectors encoding for the FRET-based sensors has not been conducted before in human myometrial cells.
Initial infection ratios were trialled based on those successfully used in neonatal mouse cardiomyocytes (see reference https://www.nature.com/articles/ncomms15031). A multiplicity of infection (MOI) of 100 was estimated to be sufficient to ensure for adequate levels of expression of the sensors. As per previous experiments in cardiomyocytes, 1 μL of virus with a titre of 1010 virions per ml applied to a cell culture of approximately 100,000 (105) cells achieved a ratio of 107 virions per 105 cells i.e., a MOI of 100 (see reference https://pubmed.ncbi.nlm.nih.gov/25783880). Further studies in neonatal rat ventricular myocytes determined that a MOI for the adenoviral vectors of 10-100 was sufficient for the sensors to be expressed at a similar fluorescence intensity in all cells (see reference https://www.nature.com/articles/ncomms15031).
An infection protocol, regularly used in cardiomyocytes, achieved successful adenoviral-mediated expression of the respective FRET reporters in the hTERT-HM cells after 3 hours of incubation with similar MOIs. The HPMCs were considerably more challenging to attain appropriate expression and localisation of the sensors initially. Different adenoviral concentrations and incubation times were trialled to achieve the optimal MOI, which was determined by the ratio between the virus titre (PFU/ml) and number of cells plated. A MOI of approximately 1000 virus particles per cell attained sufficient adenoviral transduction, which has previously been used in adult rat ventriculomyocytes (see reference https://pubmed.ncbi.nlm.nih.gov/21330599/). Due to the lower numbers of HPMCs isolated, the virus titres used for each FRET biosensor and the differing sensor expression levels, the MOI was greater than those used in cardiomyocytes for both the hTERT-HM cells and especially for the HPMCs.
We have now included the following additional information:
'For both hTERT-HM cells and HPMCs, extensive trial experiments were performed to determine the optimum concentration of virus required for adequate sensor expression and infection efficiencies. A multiplicity of infection (MOI) of approximately 1000 virus particles per cell attained sufficient adenoviral transduction, which has previously been used in adult rat ventriculomyocytes [35]. Infection efficiencies of approximately 90% were achieved for the EPAC-SH187 sensor, and 70% for the cells expressing the AKAP79-CUTie sensor.' (Lines 187-193)
' The hTERT-HM cells were incubated for 3 hours with the virus after which the media was replaced, and the cells were further incubated 18-24 hours prior to imaging of the EPAC-SH187 sensor or 48 hours for the AKAP79-CUTie sensor. Efficiency of infection was 90-100% for the EPAC-SH187 sensor and 70-80% for the AKAP79-CUTie sensor.' (Lines 174-178)
- We want to know how efficient the virus has been transduction to conduct the follow-up experiments? Could relevant evidence be provided to show that?
Please see our response to the question above in regard to virus efficiency.
- In the protein imprinting experiment, the author took GAPDH as the internal reference. Have you done similar experiments when the internal reference is β-Actin?
We have trialled multiple housekeeping genes in our previous studies (Singh et al, 2017 - see reference below). GAPDH expression has been extensively studied and has been shown to be reliable and widely expressed in human tissues. GAPDH has demonstrated stability in the use of reproductive tissues, predominantly in the placenta and myometrium making it a reliable housekeeping gene.
Singh, N., Herbert, B., Sooranna, G., Orsi, N., Edey, L., Dasgupta, T., Sooranna, S., Yellon, S., & Johnson, M. (2017). Is myometrial inflammation a cause or a consequence of term human labour?, Journal of Endocrinology, 235(1), 69-83.
Robert D. Barber, Dan W. Harmer, Robert A. Coleman, and Brian J. Clark. GAPDH as a housekeeping gene: analysis of GAPDH mRNA expression in a panel of 72 human tissues. Physiological Genomics 2005 21:3, 389-395.
Arenas-Hernandez M, Vega-Sanchez R. Housekeeping gene expression stability in reproductive tissues after mitogen stimulation. BMC Res Notes. 2013; 6:285. Published 2013 Jul 22.
Atwan, Z.W. GAPDH spike RNA as an alternative for housekeeping genes in relative gene expression assay using real-time PCR. Bull Natl Res Cent 44, 32 (2020).
- Please explain why there are seven bands in P0 generation cells and only six bands in P4 generation cells in Figure 8 C? Are the seven bands corresponding to P0 proteins extracted from seven different biopsy tissues?
For this study additional experiments were conducted using the HPMCs at passage 0 compared to passage 4. The 7 bands at P0 and 6 bands at P4 as shown in Figure 8C, F & I western blots represent protein levels from HPMCs cultured from individual donor biopsies. To avoid confusion and to make clear that for P0 we have additional samples we now indicate in each lane in Figure 8 the code for each individual donor from whom the tissue was obtained.
- The author lacks discussion on the phenomenon that the mRNA level from P0 to P4 b2-AR remains unchanged while the protein level has a downward trend as shown in Figure 8D, E and F.
At this time, we do not have an explanation for this observation and future studies will be required to confirm this finding and define the mechanism responsible.
We have however now described in the discussion the possible reasons for a decrease in the protein level relative to unchanged mRNA levels. See below:
'One possible explanation for the larger cAMP signals produced to isoproterenol with increasing passage could be due to an increase in -AR expression. The protein levels of the -AR, which is the predominant receptor subtype in term pregnant myometrium [61], were investigated with passage. Surprisingly, the opposite was seen in that there was a significant reduction in receptor expression from P0 to P4. The reasons for this decrease are unclear and further studies will be required to confirm and explain these findings. Despite a reduced protein expression, mRNA levels for -AR remain unchanged with passages. This observation could be explained by increased protein turnover or effects due to altered efficiency of mRNA translation.' (Lines 520-531)
- Explain the meaning of red, green, and blue borders in Figure 1.
This has been added to Figure 1. The green and blue borders are regions of interest drawn around the cell to monitor changes in fluorescence intensity. A region devoid of cells (red circle) is used for background correction.
- In Figure 2 and Figure 3, it is suggested to add the meaning of triangle and square.
In Figure 2 and 3, the circles are representative of hTERT-HM cells, and the square is HPMCs. Each bar is labelled accordingly.
- It needs more information about EPAC-SH187 and AKAP 79-CUTie. Like cloning or company.
The references for the two sensors are included in methods section of the manuscript. These sensors are not available commercially. See below:
'Subsequently, the cells were infected with viral vectors encoding for either a cytosolic FRET based sensor (EPAC-SH187)[33] or a version that is targeted to the plasma membrane via fusion to the scaffolding protein AKAP79 (AKAP79-CUTie)[34].' (Lines 169-172)
Reference 33: Klarenbeek, J., et al., Fourth-generation epac-based FRET sensors for cAMP feature exceptional brightness, photostability and dynamic range: characterization of dedicated sensors for FLIM, for ratiometry and with high affinity. PLoS One, 2015. 10(4): p. e0122513.
Reference 34: Surdo, N.C., et al., FRET biosensor uncovers cAMP nano-domains at β-adrenergic targets that dictate precise tuning of cardiac contractility. Nature Communications, 2017. 8(1): p. 15031.
- In the discussion part, it is suggested to add and summarize the relevant mechanism of cAMP in playing the role of delaying premature delivery.
We describe in the introduction to some extent the mechanism of cAMP promoting myometrial relaxation during pregnancy (lines 35-44 and lines 63-66). However, we have now added a paragraph to the start of the discussion section summarizing this further. See below:
'The precise cellular processes involved in the initiation of spontaneous human labour have still not been fully determined. Solving this challenge would reduce the devastating complications and adverse perinatal outcomes of PTL [4]. A complex cross-talk of hormonal, biochemical, electrical, and mechanical influences are understood to activate and stimulate the myometrium to establish uterine contractions. Myometrial cAMP signalling is upregulated during pregnancy promoting uterine quiescence [6, 8, 13, 43-45]. Altered expression in its central signalling components and a switch in effector activity, in combination with the modulation of specific pro-labour genes such as OTR, are considered to promote the fundamental switch from a relaxed uterine state to a contractile one [6, 8, 13, 14, 46, 47].' (Lines 422-431)
- The tables mentioned in Line-105 and 119 are not found in the article.
These tables (S1 to S3) were included in the supplementary material to the manuscript. We are sorry if this reviewer was not given access to this.